# Glucosylceramide Synthase Inhibitors Induce Ceramide Accumulation and Sensitize H3K27 Mutant Diffuse Midline Glioma to Irradiation

**DOI:** 10.3390/ijms24129905

**Published:** 2023-06-08

**Authors:** Khalifa El Malki, Pia Wehling, Francesca Alt, Roger Sandhoff, Sebastian Zahnreich, Arsenij Ustjanzew, Carolin Wilzius, Marc A. Brockmann, Arthur Wingerter, Alexandra Russo, Olaf Beck, Clemens Sommer, Malte Ottenhausen, Katrin B. M. Frauenknecht, Claudia Paret, Jörg Faber

**Affiliations:** 1Department of Pediatric Hematology/Oncology, Center for Pediatric and Adolescent Medicine, University Medical Center, Johannes Gutenberg-University Mainz, 55131 Mainz, Germany; 2University Cancer Center (UCT), University Medical Center, Johannes Gutenberg-University Mainz, 55131 Mainz, Germany; 3Lipid Pathobiochemistry, German Cancer Research Center, 69120 Heidelberg, Germany; 4Helmholtz-Institute for Translational Oncology Mainz (HI-TRON), 55131 Mainz, Germany; 5Department of Radiation Oncology and Radiation Therapy, University Medical Center, Johannes Gutenberg University Mainz, 55131 Mainz, Germany; 6Institute of Medical Biostatistics, Epidemiology and Informatics (IMBEI), University Medical Center, Johannes Gutenberg-University Mainz, 55131 Mainz, Germany; 7Department of Neuroradiology, University Medical Center, Johannes Gutenberg University Mainz, 55131 Mainz, Germany; 8German Cancer Consortium (DKTK), Site Frankfurt/Mainz, Germany, German Cancer Research Center (DKFZ), 69120 Heidelberg, Germany; 9Institute of Neuropathology, University Medical Center, Johannes Gutenberg-University Mainz, 55131 Mainz, Germany; 10Department of Neurosurgery, University Medical Center, Johannes Gutenberg-University Mainz, 55131 Mainz, Germany; 11National Center of Pathology (NCP), Laboratoire National de Santé, 3555 Dudelange, Luxembourg; 12Luxembourg Center of Neuropathology (LCNP), Laboratoire National de Santé, 3555 Dudelange, Luxembourg; 13Research Center of Immunotherapy (FZI), University Medical Center, Johannes Gutenberg-University Mainz, 55131 Mainz, Germany

**Keywords:** H3K27 mutant diffuse midline glioma, ependymoma, miglustat, eliglustat, glycosphingolipids, GD2, trimethylation

## Abstract

H3K27M mutant (mut) diffuse midline glioma (DMG) is a lethal cancer with no effective cure. The glycosphingolipids (GSL) metabolism is altered in these tumors and could be exploited to develop new therapies. We tested the effect of the glucosylceramide synthase inhibitors (GSI) miglustat and eliglustat on cell proliferation, alone or in combination with temozolomide or ionizing radiation. Miglustat was included in the therapy protocol of two pediatric patients. The effect of H3.3K27 trimethylation on GSL composition was analyzed in ependymoma. GSI reduced the expression of the ganglioside GD2 in a concentration and time-dependent manner and increased the expression of ceramide, ceramide 1-phosphate, sphingosine, and sphingomyelin but not of sphingosine 1-phosphate. Miglustat significantly increased the efficacy of irradiation. Treatment with miglustat according to dose recommendations for patients with Niemann–Pick disease was well tolerated with manageable toxicities. One patient showed a mixed response. In ependymoma, a high concentration of GD2 was found only in the presence of the loss of H3.3K27 trimethylation. In conclusion, treatment with miglustat and, in general, targeting GSL metabolism may offer a new therapeutic opportunity and can be administered in close proximity to radiation therapy. Alterations in H3K27 could be useful to identify patients with a deregulated GSL metabolism.

## 1. Introduction

Diffuse midline glioma (DMG) are malignant tumors of the central nervous system (CNS), occurring predominantly in childhood and associated with an exceptionally poor prognosis.

Pontine tumors with infiltrative growth have historically been subsumed under the term “Diffuse intrinsic pontine glioma” (DIPG). In 2016, resulting from an increased knowledge of pathogenetic principles, the WHO, in the fourth edition of the classification for CNS tumors, introduced the new tumor entity “DMG, H3K27M-mutant” (H3K27M-mut DMG) [1]. More than 70% of all tumors designated as DIPG carry the pathognomonic histone mutation [2,3]. More precisely, a somatic mutation in *H3F3A* or, more rarely, in *HIST1H3B* results in a lysine to methionine substitution at position 27 (H3K27M) of histone H3.3 or H3.1, respectively [3]. The H3K27M mutations lead to the hypomethylation of H3K27, promoting a more accessible chromatin state and, therefore, aberrant gene expression [4,5]. In total, 50% of H3 mutant DMGs carry a mutation in the *TP53* gene, which appears to be associated with a less favorable prognosis [6]. As other mechanisms (e.g., aberrant overexpression of EZHIP) can result in global reduction in trimethylation, the recent fifth edition of the WHO classification defines these tumors as “DMG, H3 K27-altered” [7].

Stereotactic biopsies can be performed safely and routinely nowadays [8]. This progress, along with an improved understanding of pathogenetic mechanisms, has facilitated many clinical trials. However, despite these intensive efforts to advance treatment options, radiotherapy (RT) remains the only therapeutic option with prognostic relevance to date [9]. Difficulties in the development of new therapies arise from the blood–brain barrier (BBB), which many potentially effective drugs are unable to cross [10].

Temozolomide (TMZ), an alkylating cytotoxic agent that is effective in the treatment of malignant CNS tumors (reviewed in [11]), has not shown a prognostic advantage by combination with RT in the treatment of DMGs [12,13,14]. Due to the lower toxicity of TMZ compared to other polychemotherapies [15] and a lack of alternative therapeutic options, combined chemoradiotherapy with TMZ remains in use as a therapeutic modality. However, the prognosis is consistently poor, with an overall survival (OS) of 9–11 months [16,17].

Emerging therapeutic approaches are considering gangliosides as a potential target for anti-tumor therapy. Gangliosides are amphiphilic glycosphingolipids (GSL) that carry N-acetylneuraminic acid residues [18]. They are mainly enriched in neuronal plasma membranes and change dramatically in composition and expression patterns during embryonic development and the prenatal period (reviewed in [19]). Additionally, they are associated with key processes of malignant transformation such as tumor growth, differentiation, and the ability to metastasize (reviewed in [20]). In particular, the disialoganglioside GD2 is characterized by a weak expression in normal tissue, which is predominantly localized to the CNS and peripheral nervous system (PNS) [21], and a comparatively high expression in solid tumors such as neuroblastoma [22,23] or Ewing’s sarcoma [24,25]. H3K27M-mut DMGs also exhibit strong and homogeneous expressions of GD2 [26], predisposing them to potential anti-GD2 therapy. GD2-CAR T-cell therapy for H3K27-mut DMGs is currently being tested in a clinical trial [27].

Gangliosides are components of the sphingolipid metabolism in which ceramide, which is formed in the endoplasmic reticulum (ER), is of central relevance (Figure 1). The first step in the synthesis of gangliosides is the formation of glucosylceramide (GlcCer) from ceramide catalyzed by the glucosylceramide synthase. Glucosylceramide synthase inhibitors (GSI), such as eliglustat or miglustat, are used in the treatment of lysosomal storage diseases such as Gaucher disease [28]. We have previously shown that eliglustat inhibits the proliferation of primary H3K27M-mut DMG cells [25]. However, unlike eliglustat, only miglustat can be effectively enriched in the CNS [29,30]. A study published in 2021 by Jennemann et al. also demonstrated an inhibitory effect of the GSIs miglustat and Genz-123346 (Genz) in colorectal cancer in vitro and in vivo [31].

Here, we analyzed the mechanism of action of GSIs and their therapeutic effect in H3K27M-mut DMG. Our results show that the administration of miglustat is feasible in these patients and that particularly the combination with RT should be favored in future therapy protocols. Moreover, our data suggest that losses or reductions in H3.3 K27 trimethylation can predict alterations in the GSL pathway.

## 2. Results

### 2.1. Eliglustat and Miglustat Affect the GSL Synthesis in H3K27M-Mut DMG Cells

Eliglustat and miglustat inhibit the synthesis of GSL, and, therefore, a reduction in GSL expression is expected upon incubation with either inhibitor. The ganglioside GD2 is highly expressed in H3K27M-mut DMG and, therefore, can be used as a surrogate biomarker to assess the efficacy of the inhibition on GSL synthesis. To verify whether the effect of GSI on tumor cell proliferation is accompanied by a change in the expression of GD2, the H3K27M-mut DMG SF8628 cells were treated with eliglustat or miglustat, and GD2 expression was subsequently analyzed by flow cytometry. Miglustat was used at 100 µM because previous studies have shown that the treatment with miglustat at this concentration resulted in an almost complete GSL depletion [31]. Treatment with 100 µM miglustat for 72 h did not significantly impair cell viability (Appendix A). For eliglustat, three concentrations were used: 5, 20, and 40 µM. The higher concentrations were applied only for 24 h because we have previously shown that eliglustat starts to affect H3K27M-mut DMG cell viability after 24 h of treatment [25]. No effect of 5 µM eliglustat on cell viability was observed after 72 h (Appendix A). As shown in Figure 2a, after treatment with 40 µM eliglustat over a 24 h period, the percentage of GD2 positive cells and the GD2 mean intensity were significantly reduced. Eliglustat at lower concentrations (5 µM and 20 µM) did not significantly affect the percentage of GD2 positive cells after 24 h but significantly affected the GD2 mean intensity at 20 µM. Extending the incubation period to 72 h reduced the percentage of GD2-positive cells from 87.27% (SD ± 0.91) to 76.23% (SD ± 2.72) at 5 µM (Figure 2b) and GD2 intensity. Miglustat induced a slight but significant decrease in the percentage of GD2-positive cells and in GD2 intensity after 72 h of treatment (Figure 2c). In addition, a further downregulation of GD2 with the prolongation of the treatment was observed (10 d time point, Appendix A). Other gangliosides and GSL series can be affected by GSI, including the globo- and lacto-series (Figure 1), which were indeed shown to be downregulated upon treatment with miglustat in colorectal cancer cells [31]. By mass spectrometry, we were able to show a significant reduction in five gangliosides (GM3, GM2, GM1, GD3, and GD2) and in neutral GSL after incubation with 100 µM miglustat for 72 h (Figure 2d).

Similar results were obtained with a primary cell line (409), which was isolated from a biopsy of the male patient described later in this article and which has been characterized previously [25]. Again, GSI reduced the expression of GD2 on the cell surface and of other gangliosides and neutral GSL (Appendix A).

In summary, these results indicate that both eliglustat and miglustat have an impact on GSL expression. GD2 composition changes as a function of time, and the lower concentrations are effective upon treatment prolongation.

### 2.2. Eliglustat and Miglustat Induce Accumulation of Ceramide in H3K27M-Mut DMG Cells

To assess the effect of GSI on the sphingolipid metabolism, the concentration of different components of the sphingolipid metabolism (ceramide, ceramide1-phosphate (Cer1P), sphingosine, sphingosine 1-phosphate (S1P), and sphingomyelin) was assessed by mass spectrometry (Figure 3). The treatment of SF8628 cells with 5 µM, 20 µM, and 40 µM eliglustat for 24 h resulted in concentration-dependent increases in the concentrations of ceramide and Cer1P but not S1P. Treatment with 100 µM miglustat for 72 h resulted in a significant increase in ceramide but not S1P. Cer1P was also increased, although it did not reach significance due to the high standard deviation. Sphingomyelin and sphingosine increased significantly after incubation with 5 µM, 20 µM, and 40 µM eliglustat for 24 h (Appendix A). Treatment with 100 µM miglustat for 72 h increased sphingomyelin and sphingosine, although not significantly (Appendix A). Eliglustat treatment induced high ceramide concentration also in primary H3K27M-mut DMG cells (Appendix A). In conclusion, ceramide, the substrate of glucosylceramide synthase, accumulates upon GSI treatment in a concentration-dependent manner, as shown for eliglustat. Similarly, Cer1P, a bioactive sphingolipid, which is derived by phosphorylation of ceramide, increases with increasing ceramide concentrations. However, S1P, another cell signaling molecule, which is derived from the degradation of ceramide and subsequent phosphorylation, does not alter upon GSI treatment.

### 2.3. Miglustat Has a Synergistic Effect with Ionizing Radiation

RT and TMZ are both used in the standard therapy of DMGs. We examined whether the efficacy of TMZ or RT could be improved by combining it with miglustat. As our previous results showed that GSL compositions change over time after treatment with miglustat, cells were incubated with miglustat for 72 h and then treated with TMZ after this period. DMG cells are resistant to high concentrations of TMZ (Figure 4a). A reduction in proliferation was observed after incubation with TMZ and miglustat, albeit a small one (Figure 4a). Ceramide metabolism is closely related to the cellular radiation response, and increased ceramide production can promote the radiosensitization of tumor cells. To investigate whether DMG-derived tumor cells can be radiosensitized by GSIs, the clonogenic survival of SF8628 cells was determined after concomitant treatment with X-rays and eliglustat or miglustat (Figure 4b). Treatment with 1 µM and 5 µM eliglustat or with 100 µM miglustat alone reduced the plating efficiency (PE) of SF8628 cells to 86.16% (SD ± 8.08%) and 68.78% (SD ± 15.06%) or significantly to 57.64% (SD ± 1.90%), respectively. Exposure to X-rays caused a dose-dependent reduction in the clonogenic survival of SF8628 cells. Concomitant administration of X-rays and miglustat resulted in a synergistic effect and the significant radiosensitization of SF8628 cells in an intermediate dose range of 2–4 Gray (Gy). The sensitizer enhancement ratios (SER) of 1 µM and 5 µM eliglustat or 100 µM miglustat were 1.24 and 1.48 or 2.21, respectively. In summary, while miglustat only slightly improved the efficacy of TMZ, significant radiosensitization was observed for H3K27M-mut DMG cells.

### 2.4. The Clinical Use of Miglustat in Two Patients with H3K27M-Mut DMG

Herein, we report two cases of pediatric diffuse midline glioma with H3K27M mutation. Both patients received treatment with miglustat, either alone or shortly after radiotherapy in combination with TMZ.

A seven-year-old boy was diagnosed with DMG, following complaints of headaches, and abducens nerve palsy. MRI of the brain showed a brainstem tumor without contrast medium enhancement (Figure 5). He underwent a biopsy (tumor localization on the left side), which revealed a glial tumor with significantly increased mitotic and proliferative activity and an H3K27M mutation.

Induction treatment according to the HIT-HGG-2013 protocol with RT and concomitant treatment with TMZ and valproic acid was started, followed by maintenance therapy with TMZ and valproic acid. The treatment initially resulted in a partial response. However, the child presented with progression nine months after diagnosis. He received re-irradiation and second-line oral chemotherapy with trofosfamide and etoposide. Two months after the initiation of the second-line treatment, further tumor progression combined with blood–brain-barrier dysfunction was detected.

Despite the previous multimodal treatment approach, the patient’s disease progressed rapidly. His family and the medical team decided to focus on the best supportive care. However, the family still had the desire for a potentially life-prolonging therapy that was associated with a good quality of life and that could be administered in an outpatient setting. The child’s clinical condition was still good, and, apart from the known abducens palsy and mild gait unsteadiness, there were no neurological deficits.

An individualized treatment concept was evaluated for the patient. Molecular analysis of the tumor did not identify any druggable targets. As our previous in vitro data pointed to an antineoplastic effect of the GSIs on DMG [25], and our current data indicated an accumulation of pro-apoptotic ceramide upon GSI treatment, we decided to treat him off-label with miglustat. Miglustat was selected because, unlike eliglustat, it can be effectively enriched in the CNS [33]. Informed consent was obtained from the parents. To assess the treatment efficacy of the personalized approach, we performed monthly cranial MRIs.

According to his body surface area and the recommendations for Niemann–Pick disease, he received a miglustat starting dose of 2 × 200 mg/day orally and Dexamethasone as a treatment for central edema [34]. We provided specific nutritional consultation and training to the patient and his family to prevent gastrointestinal side effects caused by suboptimal carbohydrate hydrolysis and subsequent osmotic diarrhea, which are common under treatment with miglustat [35]. After one month of therapy, we observed a regressive contrast media uptake in the left pons, particularly in the dorsal portions. However, there was also a progressive contrast enhancement towards the right cerebellum with central contrast media sparing, possibly corresponding to central necrosis or tumor progression (Figure 6a,b, arrowhead).

We increased the dose to 3 × 200 mg at good tolerability. Unfortunately, the patient developed thrombocytopenia with the higher dose of miglustat (Appendix A). This led to a dose reduction. However, to avoid hospitalization, CNS bleeding, and the necessity of regular transfusions, the therapy had to be discontinued after three months. Subsequently, the tumor rapidly progressed in areas that did not exhibit a response (Figure 6c,d) and later in multiple areas, leading to a worsening of neurological impairments. Due to the palliative setting, the patient was not hospitalized again and passed away 16 months after the initial diagnosis.

The second patient, a six-year-old girl, was transferred to our clinic from an external hospital with a history of left eye strabismus and headaches for the past 4 weeks. Further imaging (cranial MRI) showed a large partially contrast-enhancing mass in the pons. In the following biopsy, the presence of a diffuse midline glioma, H3K27M-mutant was detected. Subsequently, we initiated therapy according to the HIT-HGG 2013 protocol. Unfortunately, she had a grade II allergic reaction to valproic acid, which is part of the standard therapy in the HIT-HGG 2013 study. She had to be excluded from the study. Therefore, induction therapy was carried out with RT and TMZ, only. Routine MRI follow-up showed a response to this treatment.

No other clinical trials suitable for the patient’s overall palliative situation were available. Thus, we proposed an individualized approach exploiting the GSL metabolism of the tumor. Two months after completion of RT, we started miglustat treatment, as described for the first patient, but together with TMZ. We consulted clinical ethics and initiated supportive dietary therapy prior to the start of the therapy. In this case, as well, good clinical tolerability was observed at a starting dose of 2 × 200 mg. TMZ treatment according to the maintenance protocol for DMG treatment was applied as scheduled and the first MRI under this treatment showed an ongoing response four months after the end of radiotherapy. Similar to the first patient, she developed thrombocytopenia, and we had to reduce the dose. Especially after the TMZ pulses, thrombocytopenia was pronounced, making a dose increase in miglustat no longer possible. After another two months of low-dose miglustat and TMZ, there was a progression of the midline glioma. The patient’s parents opted for pure palliative care and a discontinuation of further therapy measures. In total, 9 months after the diagnosis, the patient died.

### 2.5. Loss of H3K27me3 May Associate with High GD2 Expression

The high expression of GD2 in H3K27M-mut DMG is driven by the H3K27M mutation, which affects the expression of the genes of the ganglioside synthesis [26]. As the consequence of the H3K27M mutation is the loss of methylation (me3) at this position, we tested also if the loss of H3K27me3 is associated with a deregulation in the ganglioside’s synthesis. We quantified the GD2 amount of five ependymoma samples and one H3K27M-mut DMG by Mass Spectrometry (Table 1). Unfortunately, tumor tissues from the two pediatric patients described in this work were not available for analysis. Therefore, tumor tissue from one adult H3K27M-mut DMG patient was used. GD2 amount in the H3K27M-mut DMG sample was not as high as in neuroblastoma [23]. GD2 expression was rather low in ependymoma; however, the only sample with loss of H3K27me3 showed a very high concentration of GD2, which was even higher than the concentration in the H327M-mut DMG tumor sample.

### 2.6. H3K27M-Mutation and H3K27-Hypomethylation-Driven Deregulation of Genes Involved in the GSL Metabolism

To analyze how H3K27 hypomethylation and H3K27M mutation affect GD2 expression, we investigated their impact on selected genes involved in GSL synthesis based on publicly available RNA-Seq datasets.

Firstly, in the gene expression analysis of DMG samples with and without the H3K27M mutation, the sample-to-sample distances revealed a relatively homogenous dataset, indicating no clear clustering pattern between the two genotype groups. (Figure 7a) Similarly, the Principal Component Analysis (PCA) of the top 1000 most variable genes did not demonstrate a meaningful pattern, with the first principal component accounting for 19.43% of the variance, and the second principal component accounting for 10.42% (Figure 7b).

Regarding specific genes, we examined the log10 normalized expression levels of the genes UGCG, ST8SIA1, and B4GALNT1, which did not show significant deregulation between genotypes, although the median expression of UGCG and ST8SIA1 tended to be higher in H3K27M mutant samples (Figure 7c). The p-values to estimate a significant differential expression between the genotype groups were assessed with a differential gene expression (DGE) analysis of the whole transcriptome and a false-discovery rate of 0.1 (results of the DGE can be found in Appendix A).

Moreover, we expanded our investigation to include posterior fossa ependymomas, focusing on the impact of H3K27 hypomethylation. Similarly, we visualized sample-to-sample distances using an expression heatmap and identified correlations between hypomethylated (blue) and hypermethylated (red) H3K27 samples (Figure 8a). The sample-to-sample distance heatmap revealed clustering based on the methylation status, although the distance values showed moderate heterogeneity between methylation groups. The PCA plot clearly differentiated between hypo- and hypermethylated ependymoma samples along the x-axis, accounting for 34.15% of the variation, and the y-axis, accounting for 22.58% of the variation (Figure 8b).

Finally, based on the results of the DGE analysis, *ST8SIA1* and *B4GALNT1* exhibited significant deregulation (*p*-adjusted values < 0.02) between the methylation groups, indicating their involvement in the methylation-driven deregulation process. *UGCG*, on the other hand, did not show significant differential expression, although two hypomethylated samples exhibited remarkably high *UGCG* expression.

These findings provide valuable insights into the impact of H3K27M mutation and H3K27 hypomethylation on gene deregulation in GSL metabolism in DIPG and posterior fossa ependymoma.

## 3. Discussion

DMG remains a tumor of unmet medical need. H3K27M-mut DMGs express a high concentration of the ganglioside GD2, which is already used as a target for CAR-T cells therapy. The high expression of GD2 suggests that the sphingolipid metabolism is particularly active in this tumor, and its modulation may affect the viability of the tumor cells. Indeed, the H3K27M mutation leads to the overexpression of genes involved in the ganglioside synthesis and particularly *B4GALNT1*, *ST8SIA1*, and *ST8SIA5* [26].

The first consequence of inhibiting the GSL pathway is the reduction in the concentration of cell surface GD2 and other gangliosides. GD2 can support tumor survival and proliferation by different mechanisms. In glioma, melanoma, and small cell lung cancer (SCLC) cells, GD2 expression has been linked to increased cell adhesion via phosphorylation of focal adhesion kinase (FAK) and adapter proteins involved in adhesion such as paxillin or p130 Cas [36,37]. The latter was more prominent in melanoma cells with GD3 positivity compared to GD2 positivity [37]. Interestingly, enhanced phosphorylation of FAK, paxillin, and p130 were also detected in osteosarcoma cells, but, here, a concomitant suppression of adhesion was observed [38]. However, these cells showed increased motility and invasiveness, which the authors attributed to the increased phosphorylation of paxillin, both by FAK and by Lyn, a kinase of the scl family [38]. The expression of GD3 and GD2 is further associated with an increased ability to proliferate and a growth advantage under serum-free or low-serum conditions [36,39,40]. In glioma cells, expression of GD3 and GD2 supported cell proliferation via the MAPK/ERK signaling pathway [36]. Gangliosides are also capable of interacting with receptor tyrosine kinases (RTK) (reviewed in [41]) as shown, for example, in breast cancer [39,40]. We cannot conclude from our experiments whether the reduction in GD2 itself contributed to the sensitization to irradiation and to the partial response observed in one of the two patients. For this, the specific inhibition of GD2 synthesis, for example, by the knockout of the GD2 Synthase (encoded by *B4GALNT1*), should be considered [26].

The second consequence of the inhibition of the GSL pathway is an increase in the concentration of ceramide, sphingosine, and Cer1P but not S1P (Figure 9). Ceramide is a mediator of apoptosis through induction of mitochondrial injury and activation of the caspase pathway and is involved in the molecular mechanisms underlying cellular response to ionizing radiation [42]. Thus, increased endogenous ceramide levels, induced by blocking the synthesis of GSL, may be responsible for the radiosensitizing effect on H3K27M-mut DMG cells and may have contributed to the partial response observed in one of the two patients. Ceramides are central to a delicate balance of different sphingolipids that can have very different effects on the death or survival of cells, especially cancer cells (reviewed in [43,44]). The interaction of the opposing components ceramide and S1P, summarized under the term “sphingolipid rheostat” (reviewed in [45]), is of high relevance. Ceramide can be hydrolyzed by ceramidases to the pro-apoptotic metabolite sphingosine. The generated sphingosine is potent in inducing apoptosis, involving reactive oxygen species (ROS) [46]. In turn, when sphingosine is phosphorylated by sphingosine kinases (SphK), S1P is produced, which has anti-apoptotic and proliferation-promoting effects [47,48,49,50]. Alkaline ceramidase 2 (ACER2) has been reported to be instrumental in the formation of the pro-apoptotic metabolite sphingosine, which also appears to be the rate-determining step in the formation of the anti-apoptotic metabolite S1P [51]. The upregulation of ACER2 depends on the presence of the wild-type (wt) p53 protein [46]. Therefore, the mutation of *TP53* in DMGs could reduce the p53-mediated degradation of ceramides to sphingosine, leading to an enhanced response to GSI through increased ceramide levels and low expression of S1P. The SF8628 cells used in this work are *TP53* mutated [52], and we did not observe a significant increase in S1P. Sphingosine was upregulated, but a comparison with a *TP53* wt cell line would be necessary to assess the effect of *TP53* mutations on sphingosine levels. *TP53* mutations occur in DMGs with a frequency of up to 50% and are associated with a worse prognosis [6], with indications that the overexpression of mutant p53 has a different effect with respect to OS than a reduction or loss of p53 [53].

While S1P was not significantly upregulated, Cer1P was. The function of Cer1P has not been fully elucidated to date. Cer1P arises from the phosphorylation of ceramide by ceramide kinase (CERK) and is a known regulator of cell survival and proliferation in a variety of cell types, including cancer cells (reviewed in [54]). Known mechanisms include interference with the PI3K/Akt signaling pathway [55,56,57], increased release of VEGF, as demonstrated in macrophages [58], and the increased phosphorylation of MAP kinases such as ERK1/2 and JNK, which is controversial [56,57]. Involvement in chemotaxis and cell migration with the participation of G-protein-coupled receptors (GPCRs) has been widely described as well [59,60,61,62]. In their review, Gomez-Larrauri et al. indicated that Cer1P-regulated reduction in pro-apoptotic ceramide formation was probably one of the main mechanisms responsible for the enhanced proliferation and cell survival under the influence of Cer1P [63]. However, as treatment of DMG cells with GSIs not only increases Cer1P levels but also causes ceramide accumulation, it can be hypothesized that the protective role of Cer1P against apoptosis may be limited or abolished in this constellation. Interestingly, other studies indicate an inhibitory role of the CERK/Cer1P on cell migration and invasion [64,65]. Altogether, this is a complex interaction of many different determinants, and the precise effects of Cer1P on the survival and functions of DMG cells treated with GSI remain unclear for the time being.

The sphingolipid metabolism is particularly active in H3K27M-mut DMG because the H3K27M mutation leads to hypomethylation of H3K27, promoting a more accessible chromatin state and, therefore, aberrant gene expression of genes involved in the ganglioside synthesis. By using GD2 as a surrogate biomarker of GSI deregulation, we observed that the loss of trimethylation at H3.3K27 in ependymoma was associated with a very high GD2 expression. The expression was even higher than the one observed in a H3K27M-mut DMG and was in the range of the expression described in neuroblastoma [23]. Loss of methylation at H3K27 in ependymoma was observed in the posteriora fossa tumors group A (PFA), which has a poor clinical outcome [66]. Thus, our data suggested that this aggressive subtype can particularly profit from a therapeutic modulation of GSL or anti-GD2 therapies. However, more samples will be necessary to confirm the association between a loss of methylation at H3K27 and high GD2 expression in ependymoma and other tumor entities. Several H3K27M-mut DMGs and some ependymoma with loss of trimethylation at H3.3K27 showed the upregulation of *UGCG*, which encodes for the first enzyme required for the synthesis of GSL and is inhibited by miglustat. Interestingly, *UGCG* is also overexpressed in other cancer types and correlates with poor prognosis [31] and with multidrug resistance protein 1 (MDR1) gene expression [67]. In ependymoma, *ST8S1A1* was downregulated, and *B4GALNT1* was upregulated in hypomethylated samples. *B4GALNT1* encodes for the key enzyme required for the synthesis of GD2, and its upregulation could explain the high GD2 level in samples with H3K27me3 reduction. A priori, one would also expect that GD3 synthase activity should be high to provide enough GD3 substrate for the synthesis of GD2. However, low *ST8S1A1* expression has been already shown in neuroblastoma with high GD2 expression and has been described as a mechanism to regulate the final ganglioside composition of a cell type (for example, high GD2 and low GD3 in the case of neuroblastoma) [68]. These data indicated that the deregulation of the GSL synthesis is complex and tumor-entity specific. The upregulation of *UGCG* and its possible role in the response to GSI inhibitors should be evaluated further.

Miglustat is an approved medication for the treatment of two rare genetic disorders, Gaucher disease and Niemann–Pick Type C disease. These diseases are characterized by the body’s inability to break down glucosylceramide. As an approved drug, the dosage and potential side effects of miglustat are well known. This makes this medication a good candidate for repurposing. Limited studies exploring the potential of GSI inhibitors in cancer treatment are available. Miglustat significantly inhibited growth rate of two different experimental mouse brain tumors, the EPEN (ependymoblastoma) and the CT-2A (poorly differentiated highly malignant anaplastic astrocytoma)[69], reduced incidence of colon cancer in an inducible colon cancer mouse model in C57Bl6 [31], and delayed tumor onset in a murine melanoma model [70]. In 2015, Ersek et al. markedly demonstrated improved osteolytic bone disease symptoms in a murine Multiple Myeloma model after treatment with miglustat [71]. The mechanism of action was mainly attributed to the depletion of GSL (especially GM3) in the proliferating tumor cells and elevated sphingomyelin-levels in non-tumor cells. Eliglustat inhibited tumor growth in a syngeneic mouse model of aggressive prostate cancer probably by inducing ceramide-mediated lethal mitophagy [72]. Interestingly, a clinical study is currently combining eliglustat with immune checkpoint inhibitors in previously treated blood and solid tumors to restore HLA-I (Human Leukocyte Antigen) antigen presentation and transform the immunogenicity of tumor cells (NCT04944888). In summary, GSI inhibitors can support anti-tumor therapy not only by directly inhibiting the tumor cells but also by remodeling their microenvironment.

Nevertheless, RT and oral chemotherapy, according to HIT-HGG 2013 protocol (TMZ and valproic acid), and re-irradiation after first progression are the only treatment options that lead to a significant extension of life expectancy in H3K27M-mut DMGs. These modalities represent the standard of care treatment in Germany and many other countries [73,74]. With respect to clinical application, the individualized regimen was given after the first-line therapy. Currently, it is not common to treat patients in first-line therapy with an experimental regimen, even if it is known that the first-line therapy will not cure the patient. In the case of progressive disease, many different approaches are under clinical evaluation. Targeting the sphingolipid metabolism is a relatively new therapeutic strategy in oncology and opens up a new field with new targets for tumor diseases. Immunotherapeutic strategies utilizing CAR-T cells and sophisticated RT techniques might be a promising opportunity [75]. One major objective for the near future will be the evaluation of the possibility of combining these different new modalities to balance the risks and benefits of each one of them. They can, indeed, be associated with severe side effects. CAR-T cell therapies, for example, pose a challenge in clinical practice. Complications, such as cytokine release syndrome, cytokine release encephalopathy syndrome, and tumor-inflammation-associated neurotoxicity, can range from mild inflammatory changes and confusion to severe symptoms such as respiratory failure, seizures, and even death [76]. Therefore, it is of great importance to define which patients could benefit from the different therapies.

## 4. Materials and Methods

*Tissues and cells.* Surplus fresh frozen tissues of ependymoma and H3K27M-mut DMG from surgery not needed for histopathological diagnosis were used for the analyses. Please refer to the sections Institutional Review Board Statement and Informed Consent Statement for details. The H3K27M-mut DMG cell line SF8628 cells were provided from ATCC (Manassas, VA, USA) and were cultured in DMEM with 5% with 10% human serum, 1% L-glutamine, and 1% penicillin-streptomycin in humidified atmosphere with 5% CO2 at 37 °C. The primary H3K27M-mut DMG cell culture 409 was isolated from a biopsy of the male patient described in this work and has been previously described [25].

*Proliferation assay.* Eliglustat and TMZ were provided by Selleckchem (Houston, TX, USA), miglustat by Biosynth (Staat, Switzerland). Stock solutions were prepared in DMSO (50 mM eliglustat and 10 mM TMZ) or in water (50 mM miglustat). In total, 3 × 10^3^ SF8628 cells were plated on 96-well plates. Cells were treated with the indicated concentrations of eliglustat, miglustat, or TMZ, respectively, for the indicated time or were mock treated. For assessing the effect of combining TMZ with miglustat, cells were pre-treated with miglustat for 72 h, the medium was changed, and simultaneous treatment with miglustat and TMZ was performed. Absorbance was measured after another 72 h of incubation. All experiments were performed three times. GraphPad Prism version 7 was used for graphical visualization and statistical analysis with student`s *t*-test. WST-1 assay (Roche, Applied Science, Penzberg, Germany) was performed according to the manufacturer’s protocol. Quantitative measurement of absorbance was performed by a microplate reader at 440 nm.

*Flow cytometric GD2 measurement.* In total, 2 × 10^6^ SF8628 or 409 cells were plated on 10 cm^2^ culture dishes and incubated with eliglustat or miglustat at the indicated concentrations or were mock treated. After trypsinization, cells were incubated with 5 µL FC BlockTM (BD Biosciences, NJ, USA) for 10 min. Subsequently, staining was performed with 5 µL anti-GD2 antibody (BD Biosciences, NJ, USA) and 10 µL 7AAD (Beckman Coulter, CA, USA) or isotype control. The pellet was then resuspended in 350 µL of PBS-BSA (Sigma-Aldrich Co., MO, USA)) and analyzed on a flow cytometer (BD Biosciences, NJ, USA). Data were visualized and analyzed using FlowJoTM version 10.8.1(BD Biosciences, NJ, USA). Visualization was in the form of histograms presented as percentages normalized to mode. All experiments were performed three times. GraphPad Prism version 7 was used for graphical visualization and statistical analysis with student’s *t*-test. Relative fluorescence intensities were calculated by dividing mean fluorescence intensities of anti-GD2 antibody-stained cells by those obtained with isotype antibody.

*Clonogenic survival.* SF8628 cells were seeded in 6-well plates with cell densities of 250, 500, 750, or 1000 cells per well for X-ray doses of 0, 2, 4, and 6 Gy, respectively. In total, 48h after seeding, the cells were treated with eliglustat (1 µM or 5 µM) or miglustat (100 µM) or were mock treated. In total, 1 h after the start of drug treatment, cells were exposed to X-rays (140 kV) at room temperature using the D3150 X-ray Therapy System (Gulmay Medical Ltd., Byfleet, UK) at a dose rate of 3.62 Gy per min or were mock treated, i.e., kept for the same time in the radiation device control room. After irradiation, the cells were incubated for another 5 days to allow colony formation and stained with 0.5% crystal violet. The drugs were kept in the medium during and after the irradiation. Colonies containing ≥ 50 cells were counted as survivors according to [77]. The plating efficiency (PE) was determined as the ratio of the number of colonies counted as survivors to the number of seeded cells. The surviving fraction (SF) was calculated as follows: SF = colonies counted/(cells seeded × PE) considering individual PE. The SER was defined as the ratio of isoeffective doses at SF 37% (D0 in radiobiology) in the absence compared to the presence of GSI. Three independent experiments were performed with biological triplicates. Data handling, plotting, and statistics were performed using SigmaPlot Version 14 (Systat Software, San Jose, CA, USA).

*Liquid-chromatography-coupled tandem mass spectrometric analysis of sphingolipids.* Liquid-chromatography-coupled tandem mass spectrometric analysis of sphingolipids was performed on an Aqcuity I class UPLC (Waters, Milford, MA, USA) coupled through electrospray ionization (ESI) to a triple-quadrupole-like tandem mass spectrometer (Xevo TQ-S, Waters, Milford, MA, USA) according to previous publications [23,78]. As described, GD2 was analyzed with an Anionoic Polar Pesticide column (5 µm, 150 mm × 2.1 mm, Waters) in MRM mode [23], whereas all other sphingolipids were separated on a reversed-phase CSH C18 column (2.1 mm × 100 mm; 1.7 μm, Waters) in MRM mode [78,79]. In positive electrospray ionization mode protonated molecular ions of S1P, Cer, Cer1P, HexCer (likely GlcCer), and Hex2Cer (likely LacCer), GM3 and GD3 ions were detected with transitions to the product ion m/z 264 and protonated molecular HexNAc-Hex2Cer (likely GA2), GM2, GD2, Tetraosylceramide (likely GA1 and Gb4Cer), and GM1 ions to the product ion m/z 204. Molecular ions of Hex3Cer (Gb3Cer) were detected by neutral loss of 504 atomic mass units. In negative electrospray ionization mode, (double)-deprotonated molecular ions of gangliosides were detected with transitions to the product ion m/z 290. All SLs were normalized with internal standards: S1P was normalized to C20-S1P, Cer1P to C12-Cer1P, ceramides to a mix of Cer(d18:1/14:0), Cer(d18:1/19:0), Cer(d18:1/25:0), and Cer(d18:1/31:0), neutral GSLs to a glucosylceramide (GlcCer) mix of GlcCer(d18:1/14:0), GlcCer(d18:1/19:0), GlcCer(d18:1/25:0), and GlcCer(d18:1/31:0). GM3 were normalized with GM3(d18:1/D5-18:0), GM2 and GD2 with GM2(d18:1/D3-18:0), GM1 with GM1(d18:1/D5-18:0), and GD3 with GD3(d18:1/D3-18:0). Data are shown as means ± SD from three independent experiments.

*RNA-Seq data analysis and visualization of publicly available data.* RNA-Seq count data of 31 High-Grade Glioma (HGG) samples of the DIPG subtype was obtained from St. Jude Cloud (https://stjude.cloud) [80]. In total, 19 samples contained a H3-3AK28M alteration. (H3K28M variant is a synonym for the H3K27M mutation.)

RNA-Seq count data of 15 posterior fossa ependymoma samples were obtained from NCBI Gene Expression Omnibus (GEO ID GSE89446) [66]. In total, 11 samples showed H3K27me3 hypomethylation and 4 samples H3K27me3 hypermethylation. After visual inspection, sample “PF17_H3K27me3 + ve” was removed as an outlier.

For each dataset, the read counts were normalized by the median of ratios method using the DESeq2 R package (version 1.34.0) [81]. Variance stabilizing transformation was performed on the data of the heatmaps and PCA plots. The decimal logarithmic transformed normalized count values with one pseudo count were used for the boxplots. The following R packages were used for the visualization: ComplexHeatmap (version 2.10.0) [82], ggplot2 (version 3.4.2) [83], and pcaExplorer (version 2.20.2) [84,85].

*Statistical analysis.* To estimate the statistical significance of the visualized genes *UGCG*, *B4GALNT1*, and *ST8SIA*, differential gene expression (DGE) analysis of the RNA-Seq data between DIPG samples containing a H3-3AK28M alteration and DIPG samples missing a H3-3AK28M alteration was performed using the DESeq2 package (version 1.34.0) by fitting the negative binomial generalized linear model for each gene and using the Wald test for significance testing. Transcripts of the count matrix with less than 10 counts in the sum of all samples were excluded. The False Discovery Rate was set to 0.1, and the model design parameter was the K28M/K27me3 status. Benjamini and Hochberg correction, as well as the “apeglm” log2 fold shrinkage method, were used to obtain *p*-adjusted values [86]. The same procedure was performed on the GSE89446 data between H3K27me3 hypomethylated and H3K27me3 hypermethylated samples. Appendix A provide both results of DGE and contain log2 fold changes, log2 fold change standard errors, and the *p* and *p*-adjusted values.

## 5. Conclusions

In conclusion, the use of miglustat in pediatric H3K27M-mut DMG patients was feasible with acceptable toxicities. In the case of the two patients described here, the therapy could also be administered shortly after RT without a relevant increase in side effects. Close observation of the blood count is mandatory, especially when miglustat is combined with conventional chemotherapy. In our case, thrombocytopenia was a dose-limiting side effect. The mechanism of action of GSI was potentially related to the increase in ceramide concentration, which is considered a mediator of apoptosis and/or to the decrease in the concentration of gangliosides such as GD2, which may support tumor survival. However, further analyses will be necessary to assess whether the change in the ceramide concentration or in the ganglioside composition directly influences apoptosis or cell proliferation of H3K27M-mut DMG cells. The balance between the components of the ceramide pathway needs to be further clarified, also in the context of the genetic background of the patients. The strong ex vivo effect observed for the combination with irradiation should be validated and needs further preclinical and clinical evaluation. Further studies are needed to show whether this treatment is also relevant for other tumor types that have H3K27 alterations such as PFA.

## Figures and Tables

**Figure 1 ijms-24-09905-f001:**
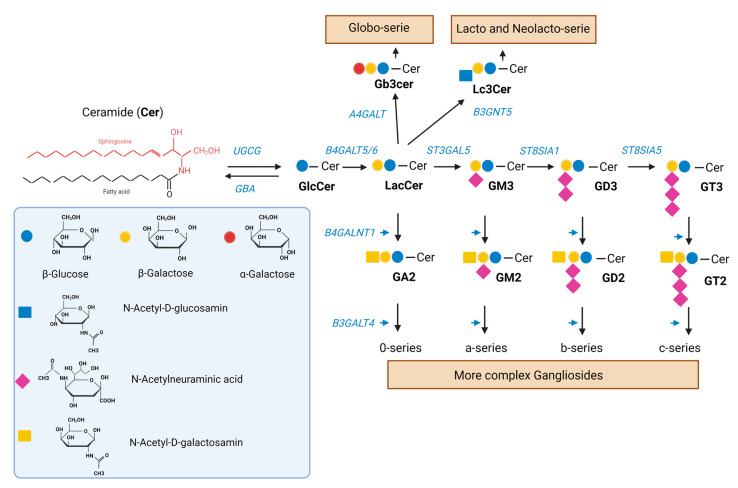
Synthesis of GSL. Ceramide (Cer) is a sphingolipid consisting of sphingosine joined by an amide linkage to a fatty acid of varying chain length and a varying degree of saturation. Glucosylceramide synthase, an enzyme encoded by the *UGCG* gene, catalyzes the synthesis of glucosylceramide (GlcCer) from ceramide. GlcCer is galactosylated to lactosylceramide (LacCer) by either of two lactosylceramide synthases, encoded by *B4GALT5/6*. LacCer is sialylated to the simple ganglioside GM3 by the sialyltransferase encoded by *ST3GAL5*. GM3 can be sequentially further sialylated by the sialyltransferases encoded by *ST8SIA1* and *ST8SIA3/5* to yield GD3 and GT3, respectively. GM3, GD3, and GT3 (as well as LacCer) can be further modified to complex gangliosides of the a-, b-, and c-series (as well as 0-series) by the enzymes encoded by *B4GALNT1* and *B3GALT4*. LacCer can be also modified to produce the globo-serie and the lacto and neolacto-serie, via the enzymes encoded by *A4GALT* and *B3GNT5*, respectively. A deficiency in the *GBA* gene encoding acidic ß-glucocerebrosidase interferes with the lysosomal degradation of glycosphingolipids and results in the accumulation of glucosylceramide mainly in macrophages [32]. GSI such as miglustat and eliglustat inhibit the glucosylceramide synthase.

**Figure 2 ijms-24-09905-f002:**
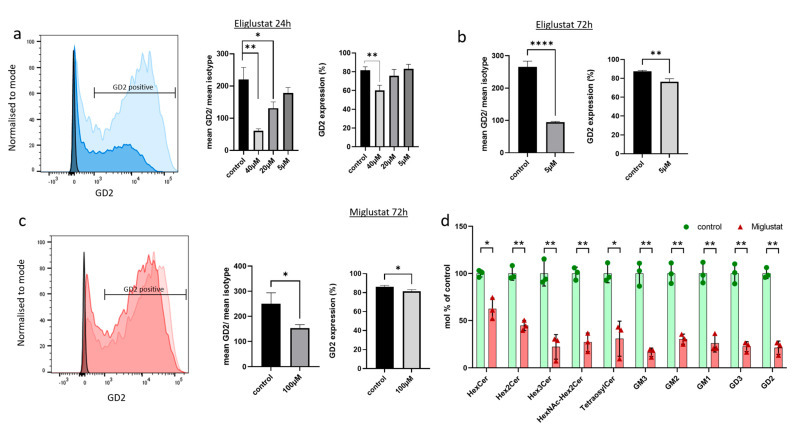
Glucosylceramide synthase inhibitors (GSIs) affect expression of GSL. (**a**–**c**) SF8628 cells were treated with eliglustat or miglustat or mock-treated (control), and GD2 surface expression was analyzed by flow cytometry. The light blue and light red histograms show GD2 expression of mock-treated cells, whereas the dark blue and dark red histograms show GD2 expression after GSI treatment (40 µM eliglustat for 24 h in (**a**) and 100µM miglustat for 72 h in (**c**)). Isotype controls are also illustrated (black histogram). One out of three independent experiments is shown. The bar graphs show the percentage of GD2-positive cells (GD2 expression) and the relative GD2 intensity (mean GD2/mean isotype) upon treatment with three concentrations of eliglustat for 24 h (**a**), with 5 µM eliglustat for 72 h (**b**), and with 100 µM miglustat for 72 h (**c**). In (**d**), SF8628 cells were treated with 100 µM miglustat or mock-treated (control) for 72 h. Glycosphingolipids were analyzed by mass spectrometry and quantified as mol percent of mock-treated cells. Following GSLs were detected: HexCer (likely GlcCer), Hex2Cer (likely LacCer), GM3, GD3, HexNAc-Hex2Cer (likely GA2), GM2, GD2, Tetraosylceramide (likely GA1 and Gb4Cer), GM1, and Hex3Cer (Gb3Cer). All GSLs were normalized with internal standards. Data are shown as means ± SD from three independent experiments. The statistical comparisons against the mock-treated samples for each dose were performed using student’s *t*-test: **** *p* < 0.0001; ** *p* < 0.005; * *p* < 0.05.

**Figure 3 ijms-24-09905-f003:**
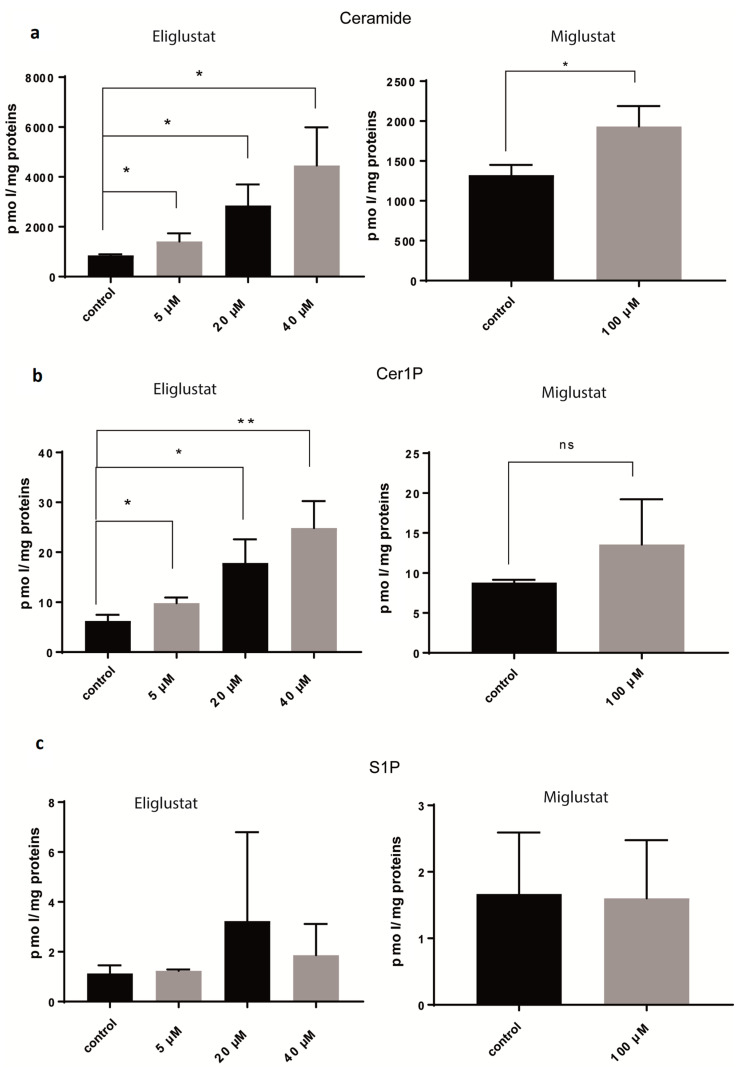
Glucosylceramide synthase inhibitors (GSIs) affect several steps of sphingolipid metabolism. SF8628 cells were treated with eliglustat at different concentrations for 24 h or with 100 µM miglustat for 72 h or mock-treated (control). Ceramide (**a**), Cer1P (**b**), and S1P (**c**) were analyzed by mass spectrometry and quantified as pmol per mg of proteins. Data are shown as means ± SD from three independent experiments. The statistical comparisons against the mock-treated samples for each dose were performed using student’s *t*-test: * *p* < 0.05, ** *p* < 0.01, ns: not significant.

**Figure 4 ijms-24-09905-f004:**
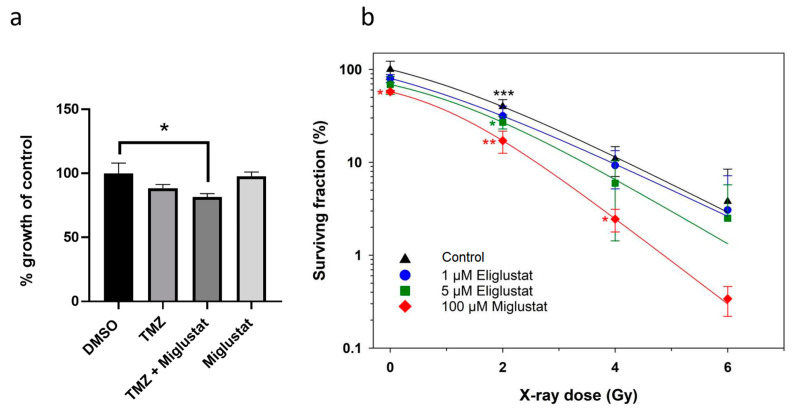
Miglustat sensitizes DMG cells to ionizing radiation. (**a**) To assess the combined effect of miglustat and TMZ, SF8628 cells were incubated with 100 µM miglustat; after three days, 10 µM TMZ was added (TMZ + Miglustat), and the incubation was continued for three days. Following controls were used: SF8628 cells incubated in medium alone for three days and subsequently incubated with either DMSO or, alternatively, 10 µM TMZ for another three days; SF8628 cells treated with 100 µM miglustat alone for 6 days. Absorbance is plotted as a percentage relative to the growth of DMSO treated SF8628 cells. Bars: mean value in % ± SD. The statistical comparison against the DMSO treated samples was performed using student’s *t*-test: * *p* < 0.05. (**b**) Combination effects of X-rays and eliglustat or miglustat on clonogenic survival. SF8628 cells were incubated with eliglustat or miglustat or mock-treated (control). Data are shown as means ± SD from three independent experiments with biological triplicates. All data were normalized to unirradiated and untreated cells. The statistical comparisons against the untreated samples for each dose were performed using student’s *t*-test: * *p* < 0.05, ** *p* < 0.01, *** *p* < 0.001. All lines were fitted by a linear-quadratic function.

**Figure 5 ijms-24-09905-f005:**
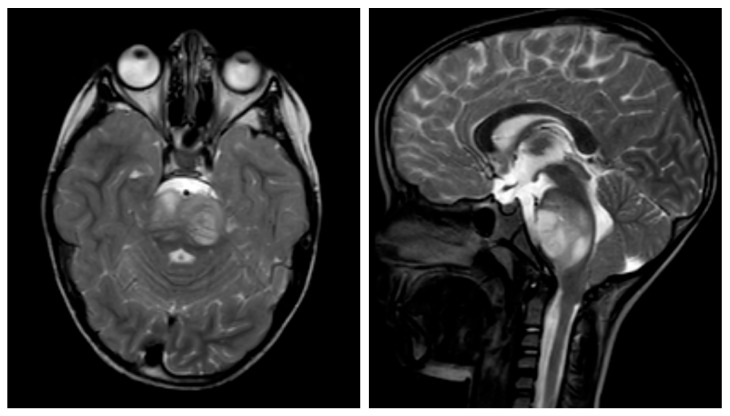
Sagittal and transverse T2-weighted turbo spin-echo (TSE) cranial magnetic resonance imaging (MRI) demonstrating diffuse midline glioma of a seven-year-old boy. The lesion appears as a mixed hyperintense and isointense area in the dorsal aspect of the brainstem, with associated edema and compression of the fourth ventricle.

**Figure 6 ijms-24-09905-f006:**
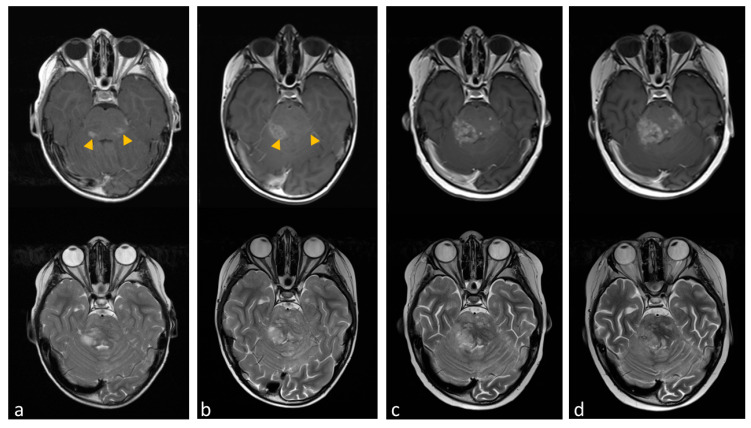
Transverse cranial magnetic resonance imaging (MRI) with T2-weighted turbo spin-echo (TSE) (lower line) and T1-weighted with gadolinium contrast media (upper line). (**a**) The image shows contrast-enhanced lesions in the pons, particularly in the dorsal portions (yellow arrowheads). (**b**) Tumor region after one month of miglustat therapy with regressive contrast media uptake in the left pons and progressive contrast enhancement towards the right cerebellum. (**c**,**d**) show tumor progression after dose reduction and later discontinuation of miglustat.

**Figure 7 ijms-24-09905-f007:**
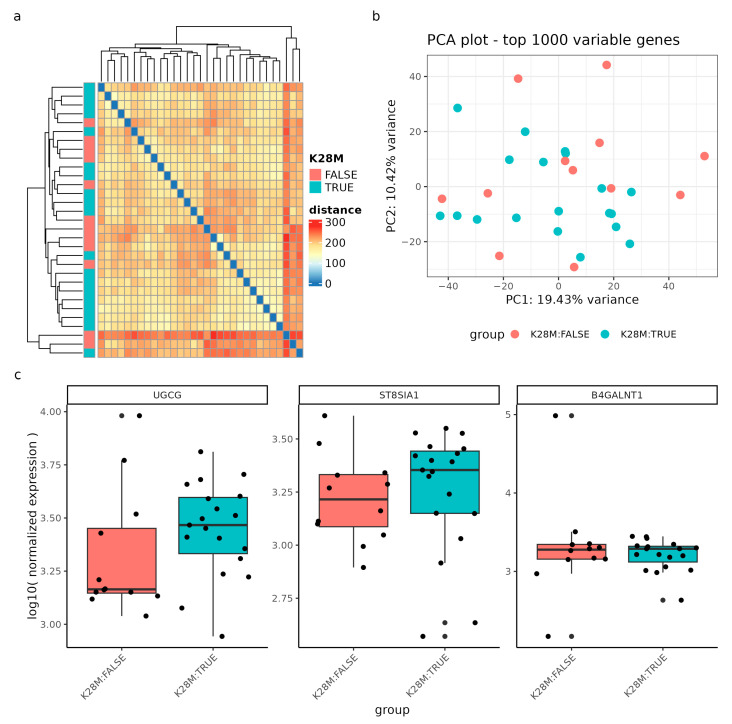
Gene expression analysis of DIPG samples (n = 31) with and without the H3K28M mutation. (**a**) Sample-to-sample distances are represented as an expression heatmap using the variance-stabilized count data. The intensity of square colors and the hierarchical clustering provide a summary of the correlations between H3K28M altered (blue) and not altered (red) genotypes. (**b**) Gene expression variation among the DIPG dataset, based on the top 1000 variably expressed genes, is displayed in a Principal Component Analysis (PCA) scatter plot. The percentages on each axis correspond to the proportions of variance that the primary components can account for. Samples are colored by the respective genotype: H3K28M altered samples (blue) and samples not containing the H3K28M mutation (red). (**c**) The log10 normalized expression of the genes *UGCG*, *ST8SIA1*, and *B4GALNT1* grouped by the genotype. Note, the nomenclature H3K28M is a synonym for the H3K27M mutation.

**Figure 8 ijms-24-09905-f008:**
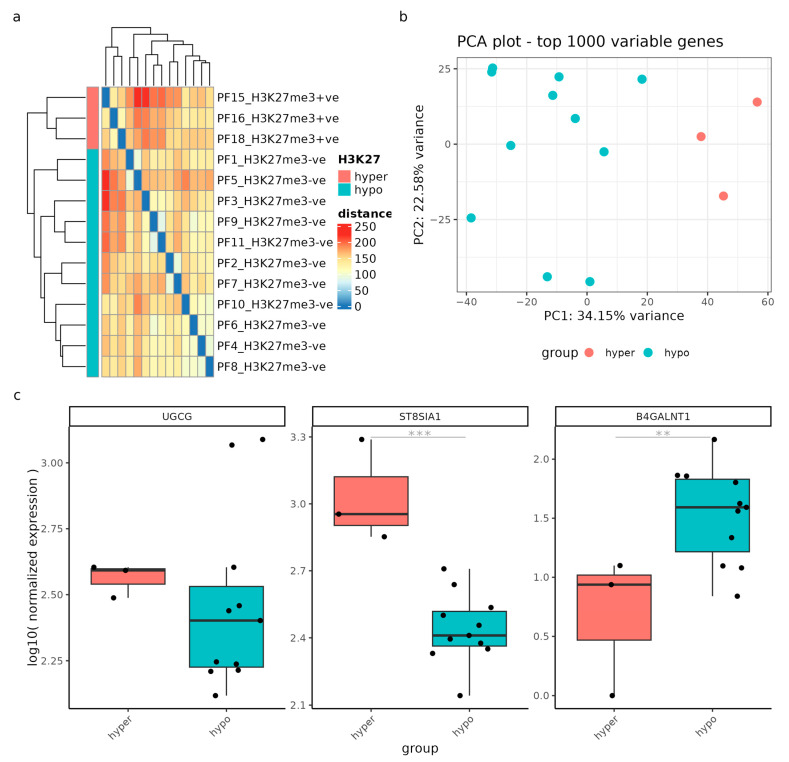
Gene expression analysis was performed on samples from patients with posterior fossa ependymomas (n = 15) to compare those with a low H3K27me3 (hypo) and high H3K27me3 (hyper). (**a**) Sample-to-sample distances were visualized using an expression heatmap constructed from variance-stabilized count data. The heatmap’s color intensity and hierarchical clustering provide a concise representation of the correlations between hypomethylated (blue) and hypermethylated (red) H3K27 samples. (**b**) The PCA scatter plot showcases gene expression variation within the GSE89446 dataset, based on the top 1000 variably expressed genes. The percentages displayed on each axis indicate the proportion of variance accounted for by the primary components. Samples are colored by hypomethylation (blue) and hypermethylation (red). (**c**) Gene expression levels of *UGCG*, *ST8SIA1*, and *B4GALNT1* are presented as log10 normalized values, categorized according to the methylation status. ** = *p*-adjusted value < 0.05; *** = *p*-adjusted value < 0.01.

**Figure 9 ijms-24-09905-f009:**
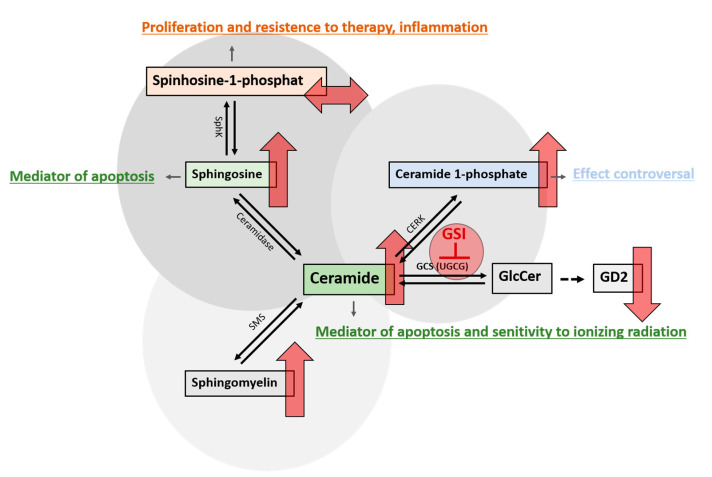
Alterations in sphingolipid metabolism after treatment with GSI. Various enzymes are responsible for the conversion of ceramide into its metabolites, which collectively determine the survival or demise of cells. Blocking the glucosylceramide synthase (GCS), which is encoded by the *UGCG* gene, with glucosylceramide synthase inhibitors (GSI), leads to an inhibition of glucosylceramide (GlcCer) synthesis. GlcCer is the precursor for the synthesis of gangliosides such as GD2, whose concentration is reduced by using GSI. The concomitant increase in pro-apoptotic ceramide leads to further changes in sphingolipid metabolism. Ceramidase can convert ceramide to sphingosine, which, in turn, can be phosphorylated by sphingosine kinase (SphK) to sphingosine 1-phosphate (S1P). GSI treatment increased sphingosine, which is known to have pro-apoptotic properties, but not S1P, a protective factor known to have proliferative effects on cells. This constellation might be influenced by the *TP53* mutational status. Another consequence is the increase in ceramide 1-phosphate formed by ceramide kinases (CERK). The predominantly anti-apoptotic effect of C1P is explained, in part, by the elimination of ceramide associated with CERK activity. However, as precisely this mechanism is prevented by the present high ceramide levels, the anti-apoptotic effect of C1P may be reduced or suppressed by GSI treatment. Additionally, the concentration of sphingomyelin produced by sphingomyelin synthase (SMS) increases as a consequence of the inhibition of GCS. The upward red arrows indicate an increase and the downward red arrows indicate a reduction in the concentration of the respective metabolite after treatment with GSI. The right left arrow shows that no changes in the concentration were observed.

**Table 1 ijms-24-09905-t001:** Expression of GD2 in tumor tissues.

Sample	Sex/Age	Diagnosis	GD2 (nmol/mg Protein)
**#1**	m/14	Ependymoma	0.156
**#2**	f/4	Ependymoma, RELA-Fusion positive	0.086
**#3**	f/1	Ependymoma, posteriora fossa, with loss of methylation at H3K27	1.624
**#4**	m/10	Anaplastic Ependymoma, RELA-fusion positive	0.128
**#5**	m/12	Anaplastic Ependymoma, RELA-fusion positive. (Relapse of Sample #4)	0.162
**#7**	f/46	H3K27M-mut DMG	0.304

## Data Availability

Data supporting the reported results can be obtained from the corresponding author. Following St. Jude Cloud datasets were used: Pediatric Cancer Genome Project (PCGP). This study makes use of data generated by the St. Jude Children’s Research Hospital—Washington University Pediatric Cancer Genome Project and/or Childhood Solid Tumor Network [87]. Access to this data has to be requested at https://platform.stjude.cloud/data/cohorts/pediatric-cancer (accessed on 10 May 2023).

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
