# Peer review of "Glucosylceramide Synthase Inhibitors Induce Ceramide Accumulation and Sensitize H3K27 Mutant Diffuse Midline Glioma to Irradiation"

_ijms, 2023, doi:10.3390/ijms24129905_

Round 1
Reviewer 1 Report
This work by El Malki et al. studied the effects of two glucosylceramide synthase inhibitors on ceramide metabolism in an H3K27-mut DMG cell line and the anti-tumor efficacy of the drugs in combination with the current first-line therapy. Then, the authors presented two clinical cases, in which the pediatric patients were treated with miglustat after they failed to respond to the standard treatment. The work is overall interesting and important to the field as it tackles a critical issue and bridges clinical cases with in vitro results. The manuscript is nicely written, and references are discussed well. However, some experiments need proper controls, and the methods could be better described. These issues should be addressed before publication.
Major comments:
1. The authors used one H3K27M-mut DMG cell line in all experiments. The reduction of GD2 and the inhibition of cell growth were quite minimal. The authors should check the effects of eliglustat and miglustat in another cell line.
2. As mentioned above, the reduction of GD2 in SF8628 cells treated with 100 uM miglustat was small. However, Jennemann et al. reported a complete depletion of the neutral glycosphingolipids and lactosylceramide after a 4-day treatment in colorectal cancer cells. Have the authors checked the expression of these lipids? Was the difference depending on cell type?
3. In figure 4a, the SF8628 cells were treated with 10 uM TMZ or 100 uM miglustat alone or in combination. Although the authors mentioned that the cells were resistant to TMZ, they still should add a vehicle control for TMZ and maybe for miglustat if the solvent was different.
4. In the experiment where the combined effects of X-rays and GSIs were evaluated, were the drugs kept in the medium during and after the irradiation?
5. In the manuscript, the authors suggested that the loss of H3K27 methylation was associated with elevated GD2 expression and could serve as a biomarker to select patients with GSL abnormalities. However, the correlation indicated by table 1 was weak. Did the authors measure the H3K27me3 level in tumor tissues from these patients? If so, the authors should plot the GD2 level versus the H3K27me3 level.
Minor comments:
1. In line 59, should it be a lysine to methionine substitution?
2. In line 191, the percentage of 6.78% should be about 67%.
3. In line 213, the authors mentioned that a biopsy was taken from the first patient, and an H3K27M mutation was detected. Did the authors record which pon the biopsy was taken from, left or right? It is interesting to speculate if there is a difference in H3K27me3 level between the two lesions, which might cause the response discrepancy to the miglustat treatment.
Author Response
Major comments:
- The authors used one H3K27M-mut DMG cell line in all experiments. The reduction of GD2 and the inhibition of cell growth were quite minimal. The authors should check the effects of eliglustat and miglustat in another cell line.
The first experiments were conducted with primary cells (nr 409) isolated from a biopsy of the male patient treated with miglustat described in this manuscript. Because few cells were available, the commercially available cell line SF8628 was used to validate the results. We have now added the results showing the effect of GSI on the expression of GD2, ceramide and GSL in the 409 cells (that were characterised in our prevoius publication PMID: 33572900, also concernig the sensitivity to GSI treatement) This is particularly relevant, because it shows that the GSI induce increase in cerammide in the patient that was treated with miglustat
- As mentioned above, the reduction of GD2 in SF8628 cells treated with 100 uM miglustat was small. However, Jennemann et al. reported a complete depletion of the neutral glycosphingolipids and lactosylceramide after a 4-day treatment in colorectal cancer cells. Have the authors checked the expression of these lipids? Was the difference depending on cell type?
Our data suggest that the incubation time has an effect on the depletion. Jennemann et al used 4-day treatment, while we used 3 days. In the revised manuscript we measured also neutral GSL and several gangliosides by mass spectrometry (Figure 1d). All were significantly reduced after treatement with 100 µM miglustat
- In figure 4a, the SF8628 cells were treated with 10 uM TMZ or 100 uM miglustat alone or in combination. Although the authors mentioned that the cells were resistant to TMZ, they still should add a vehicle control for TMZ and maybe for miglustat if the solvent was different.
We have added the vehicle control (DMSO) in Figure 4a. Miglustat was dissolved in water
- In the experiment where the combined effects of X-rays and GSIs were evaluated, were the drugs kept in the medium during and after the irradiation?
Yes, the drugs were kept in medium during and after the irradiation. We have added this in the protocol in material and methods
- In the manuscript, the authors suggested that the loss of H3K27 methylation was associated with elevated GD2 expression and could serve as a biomarker to select patients with GSL abnormalities. However, the correlation indicated by table 1 was weak. Did the authors measure the H3K27me3 level in tumor tissues from these patients? If so, the authors should plot the GD2 level versus the H3K27me3 level.
We did not analyse the level of H3K27me3 here. However, in the revised manuscript we used available RNAseq data from Posteriora Fossa Ependymoma with low and high H3K27me3 to analyse the expression of genes involved in the gangliosides Synthesis (Figure 8). Samples with low methylation have a significan higher expression of B4GALNT1, which is the key enzyme in the synthesis of GD2
Minor comments:
- In line 59, should it be a lysine to methionine substitution?
The reviewer is absolutely right. We are very sorry for this error and have changed it
- In line 191, the percentage of 6.78% should be about 67%.
We are very sorry for this error and have changed it
- 3. In line 213, the authors mentioned that a biopsy was taken from the first patient, and an H3K27M mutation was detected. Did the authors record which pon the biopsy was taken from, left or right? It is interesting to speculate if there is a difference in H3K27me3 level between the two lesions, which might cause the response discrepancy to the miglustat treatment.
This is indeed a very interesting point. In general, tumor heterogeneity is a major obstacle in the treatment of solid tumors. The biopsy in this case was taken from the left side (we added this information at line 248) and contained multiple punch biopsies (according to the surgical report ten different positions) with comparable results. One could speculate if a different tumor biology in different regions might be a reason for the different response to the miglustat treatment. Because of the vulnerable localization of the tumor and the risk of complications in association with invasive procedures, it is understandable that it’s not possible to take biopsies from both regions. However, spatial and temporal conservation of the H3K27M mutation has been described in H3K27M-mut DMG and is in contrasts with the significant heterogeneity of driver mutations observed in adult HGG (PMID: 26727948).
Reviewer 2 Report
Overview
The submitted manuscript by Malki et.al describes cell based and clinical studies with the glucosylceramide synthase inhibitors (GSIs) Eliglustat and Miglustat in diffuse midline gliomas (DMG). Whilst there is interest in DMG owing to the lack of treatment options and dismal prognosis, the cell-based studies presented in this manuscript are very simplistic and basic in nature, and minimal in quantity. Furthermore, the ability of Eliglustat to attenuate cell proliferation through interfering with gangliosides biosynthesis have been previously described by the author (Ref 25) and therefore the results from the cell-based studies with Miglustat were somewhat expected. The interesting aspect of the paper was the clinical studies in two DMG patients, but unfortunately there wasn’t a positive outcome to report from these cases and owing to the very small treatment cohort and lack of post-treatment analysis of patient samples it is hard to draw any conclusions, and thus these findings were merely observational.
Concerns and comments regarding figures
Figure 1: Whilst this is useful its not conventional to have a figure schematic in the introduction. Consider placing this into the supplementary section.
Figure 2. The gates/regions showing the % GD2 expression should be indicated in the histograms. I believe the better analysis here would be mean GD2 signal intensity calculation which is likely to provide a greater difference between the treated and the untreated sample.
Figure 3. Analysis of some of the other gangliosides that are expected to go down after Eliglustat/Miglustat would have strengthen this section. I know GD2 was shown in Figure 1 but what about some of the other gangliosides highlighted in Figure 1? For example, GM3, GT3, GT2?
Figure 4: Sensitisation enhancement ratio (SER) should be calculated for the GSIs
General comments
1. The figures and figure legends lack key detail. For example, Figure 2: what cell line, Figure 4: how long were the cells treated with TMZ and what is the label on the y-axis in Figure 4(a).
2. Although there was no notable clinical benefit of using Miglustat in the two DMG patients, and I appreciate it was not possible to analyse biopsies from these patients’ following drug treatment, some in silico analysis of key genes, (as highlighted in Figure 1), e.g., expression level/mutation status would have provided some indication of the frequency of ganglioside biosynthesis pathway deregulation in DMG and hence the potential therapeutic opportunity.
OK. No concerns.
Author Response
Concerns and comments regarding figures
Figure 1: Whilst this is useful its not conventional to have a figure schematic in the introduction. Consider placing this into the supplementary section.
We think that the figure is important to understand the interaction between different GSLs and the enzymes required for their synthesis. In the revised papers we have added additional data on other GSLs (Figure 1d) and on the expression of genes involved in their synthesis (Figure 7 and 8), and therefore we think that it would be better for the readers to have this figure in the main text
Figure 2. The gates/regions showing the % GD2 expression should be indicated in the histograms. I believe the better analysis here would be mean GD2 signal intensity calculation which is likely to provide a greater difference between the treated and the untreated sample.
Thank you for the suggestion. We have added the gates and the mean GD2 signal Intensity in the Figure.
Figure 3. Analysis of some of the other gangliosides that are expected to go down after Eliglustat/Miglustat would have strengthen this section. I know GD2 was shown in Figure 1 but what about some of the other gangliosides highlighted in Figure 1? For example, GM3, GT3, GT2?
We have added if Figure 1d the analysis of other gangliosides and other GSL after incubation with miglustat. All were significantly reduced
Figure 4: Sensitisation enhancement ratio (SER) should be calculated for the GSIs
We have added the SER value in the results and the way how it was calculated in the MM section
General comments
- The figures and figure legends lack key detail. For example, Figure 2: what cell line, Figure 4: how long were the cells treated with TMZ and what is the label on the y-axis in Figure 4(a).
We apologise and we have added the relevant details and improved the legends.
- Although there was no notable clinical benefit of using Miglustat in the two DMG patients, and I appreciate it was not possible to analyse biopsies from these patients’ following drug treatment, some in silico analysis of key genes, (as highlighted in Figure 1), e.g., expression level/mutation status would have provided some indication of the frequency of ganglioside biosynthesis pathway deregulation in DMG and hence the potential therapeutic opportunity.
We have used available RNAseq data to analyse the expression of UGCG and of the two genes required for the synthesis of GD2 in H3K27M-mut DMG mutated vs not mutated and in posteriora fossa ependymoma (low vs high H3K27me3). The results are shown in Figure 7 and 8. Ependymoma with low methylation have a higher expression of B4GALNT1, which is the key enzyme in the synthesis of GD2. H3K27M-mut DMG have more UGCG and ST8SIA1. The results are discussed from line 500
Reviewer 3 Report
In the present study, Malki et al. reported the use of miglustat in treating pediatric H3K27M-mut DMG patients. They found that GSIs have effects on the sphingolipid metabolism and induce accumulation of ceramide. They further showed that miglustat sensitizes DMG cells to ionizing radiation. This is a clinical tumor biological study with interesting observations. However, there are several shortcomings to be addressed prior to publication as below:
1. Figure 2: Have the authors analyzed cell viability under the treatment?
2. It will increase the significance of the paper to add a summary of mouse models related to using miglustat in tumor treatment and possible molecular mechanisms.
Author Response
- Figure 2: Have the authors analyzed cell viability under the treatment?
We have added the effect of 5 µM eliglustat and 100 µM miglustat for 72 hours in Supplementary Figure 1. No effect on viability was observed. The effect of higher concentration of eliglustat on cell viability can be observed only after 24 h of treatment (we have shown this in Wingerter et al, Cancers 2021 ) and therefore we did not use longer incubations. We have commented this in the results section (from line 136)
- It will increase the significance of the paper to add a summary of mouse models related to using miglustat in tumor treatment and possible molecular mechanisms.
We have added the summary in the discussion from line 519
Round 2
Reviewer 1 Report
The authors have addressed most of the reviewers’ comments and added sufficient data to support their conclusions. The authors also better described the methods and discussed the limitations. The manuscript has been improved and is suitable for publication.
Author Response
We thank the reviewer for the positive feedback. We have improved the conclusions by clarifying that the mechanism of action of miglustat has still to be disclosed, particularly the effect of ceramide concentration on apoptosis and of the ganglioside composition of on cell proliferation in DMG cells
Reviewer 2 Report
The authors have addressed the majority of comments and concerns raised to an acceptable standard. The addition of new data as well as analysis and clarification of previous data makes this article more suitable for publication.
No major issues or concerns.
Author Response
We thank the reviewer for the positive feedback. We improved the description of the methods by clarifying when the mock-control was used in the legends of figure 2, 3 and 4 and supplemental S2 and S3 and in the material and methods section. We reorganized the description of Figure 4, and we corrected Figure 3 and S4 (the quantification was always done as pmol per mg of proteins). We also corrected wording errors